# The Impact of ESG Scores on Bank Market Value? Evidence from the U.S. Banking Industry

Ersan Ersoy [1,*], Beata Swiecka [2], Simon Grima [3,4], Ercan Özen [1] and Inna Romanova [4]

1   Faculty of Applied Sciences, Usak University, Uşak 64000, Turkey; ercan.ozen@usak.edu.tr
2   Institute of Economics and Finance, University of Szczecin, 70-453 Szczecin, Poland;
    beata.swiecka@usz.edu.pl
3   Department of Insurance and Risk Management, Faculty of Economics, Management & Accountancy,
    University of Malta, MSD2080 Msida, Malta; simon.grima@um.edu.mt
4   Faculty of Business, Economics and Management, University of Latvia, LV-1586 Riga, Latvia;
    inna.romanova@lu.lv
*   Correspondence: ersan.ersoy@usak.edu.tr

**Abstract:** Although there is a large volume of literature on the relationship between Environmental, Social and Governance (ESG) and firm performance, only a limited number of studies have focused on the banking sector. In addition, most of them used linear models. Therefore, in this study, we examined the impact of ESG and ESG pillar scores (environmental, social, and governance) on the market value of U.S. commercial banks by using linear and non-linear panel regression models over the period of 2016–2020. Moreover, we used the market value as a bank value indicator and included the effect of COVID-19. Results show an inverted U-shaped relationship between market value and ESG and The Social Pillar Score (SPS) and a U-shaped relationship between market value and The Environment Pillar Score (EPS). Findings from this study are important indicators for investment managers and policymakers who want to maximise bank market value while complying with ESG standards.

**Keywords:** ESG; bank performance; bank value; sustainability; COVID-19





## 1. Introduction

Investment performance is being measured and analysed according to sustainability criteria measurements based on corporate environmental, social and governance (ESG) scores. This ESG score combines social needs, economic criteria and the environment, allowing the investors to determine companies' sustainable performance and low-risk investment opportunities.

Refinitiv's ESG scores measure a firm's relative ESG performance, commitment, and effectiveness based on firm-reported data. The ESG pillar score has 10 categories; the ESG combined score has 3: Environmental, Social, and Governance Pillar Scores. The Environment Pillar Score (EPS) measures resource use (water, energy, sustainable packaging, and environmental supply chain), emissions (emissions, waste, biodiversity, and environmental management system), and innovation (product innovation, and green revenues, research and development and capital expenditures). The Social Pillar Score (SPS) includes workforce (diversity, inclusion, working conditions, health and safety), human rights, community, and product responsibility (responsible marketing, product quality, data privacy). The Governance Pillar Score (GPS) includes management structure (independence, diversity, committees, and compensation), shareholders (shareholder rights and takeover defences), and Corporate Social responsibility (CSR) strategy (CSR strategy, ESG reporting and transparency) [1].

Banking, which is an important pillar in the financial sector, has an important role in sustainable development. Therefore, since sustainability is one of the most vital trends

in the banking industry, investors need to ensure sustainable and responsible investing by analysing corporate social responsibility, corporate governance structures and environmental issues when making investment decisions [2]. This is also due to the fact that sustainability reporting is gaining significant acceptance worldwide as stakeholders feel the need for greater transparency on environmental, social and governance (ESG) issues [3].

Firms' social, environmental, and governance (ESG) practices are important for all stakeholders. Therefore, the impact of ESG on firm performance and firm value is a topic of increasing importance in literature. Several theoretical and empirical studies exist on the relationship between a firm's ESG and financial performance. According to neoclassical theory, ESG performance negatively affects financial performance. Based on the neoclassical theory, spending on environmental and social causes relates to a competitive disadvantage and cost increase [4]. The stakeholder theory stated that a firm's main purpose should be to maximise shareholder value and create value for all stakeholders, such as employees, consumers, and natural or environmental resources [5]. This theory suggests that shifting from shareholder-focused to stakeholder-focused governance would balance the interests of investing and non-investing stakeholders in banks, thereby protecting management's excessive risk-taking and bank value [6].

According to the stakeholder and legitimacy theory, a bank's socially responsible investment positively affects the firm's performance. In accordance with the stakeholder and resource-based theories, environmental investment has a positive influence. Based on the agency theory, better corporate governance will improve bank performance since, in the banking industry, better corporate governance disclosures are needed to eliminate conflicts of interest between managers and shareholders and reduce the agency problem [7].

Most previous studies investigating the effect of ESG on bank value and performance have assumed that the relationship is linear. However, findings from theoretical approaches and empirical studies demonstrate that there may not be a linear association between the ESG and bank value. For example, corporate governance provides accountability, compliance, transparency, and decreased agency costs for financial stakeholders. On the other hand, it is observed that responsible practices such as quality, safety, diversity and equal opportunity in employment, attention to human rights, quality and safety in products and services decrease the firm's market value. For this reason, the impact of ESG performance on bank value is complex [5]. In addition, managers are willing to invest in socially responsible investments since this can be a strategic tool that brings competitive advantages and increases social welfare. This win-win strategy encourages managers to keep investing in socially responsible investments. However, there may be important risks that arise in practice. Socially responsible investments lead to decreased firms' resources due to financial allocations, human resources, and managerial inputs. In addition, continuous investment in socially responsible investments can lead to more resource competition between different departments. For this reason, the impact of socially responsible investments on the company's performance may turn negative after a certain point. Therefore, it may cause a non-linear rather than linear association [8]. El Khoury [7] suggested that the costs of ESG investments exceed the benefits in the long term, and therefore the impact of ESG activities on the market performance of banks may be changed.

The aim of this study is to investigate whether ESG and ESG pillar scores affect the bank market value for 176 U.S. commercial banks from 2016 to 2020. We have also examined the impact of bank-specific variables and the COVID-19 pandemic on market value. This paper contributes to the literature in several ways. There are several studies on ESG and firm performance, but only a limited number of studies have focused on the banking sector. The paper empirically examines the impact of ESG and ESG pillar scores on the market value of U.S. commercial banks. Thus, this study has extended the existing literature by examining the banking sector. Previous studies used Return on Assets (ROA), Return on Equity (ROE), market to book value, stock return and Tobin's Q etc. as performance indicators. Unlike these studies, the bank market value was used as a performance indicator in this study. As far as we know, this is the first study that examines the impacts of ESG

performance on the bank market value. Thereby, this study brings new insights to the sustainability literature. Additionally, linear models have generally studied the relationship between ESG and firm performance. However, we have examined the impact of ESG on the bank market value using linear and non-linear models. This study provides empirical evidence that an inverted U-shaped relationship exists between bank market value and ESG and SPS and U-shaped relationship between bank market value and EPS. Moreover, it also took into account the effect of COVID-19.

This paper provides important practical implications for the banking industry, managers and other stakeholders. Our findings show a non-linear association between ESG scores and bank market value. U.S. Managers who want to maximise bank market value should consider the inverted U-shaped relationship between market value, ESG and SPS and the U-shaped relationship between market value and EPS. Besides, the empirical results of this paper help all stakeholders and regulators better understand the effect of ESG performance on bank market value. Considering the non-linear relationship between ESG performance and market value, it should be determined why the positive effect of ESG on market value turns negative. Thus, it will be possible to take measures to reduce the negative effects on market value. The development of policies that will make social, corporate governance and environmental activities projects more visible by managers may increase the demand of environmentally-conscious, socially responsible customers and investors for bank's services and stocks. At the same time, managers can gain competitive advantages for their banks through more visible ESG performance disclosure.

The rest of the paper is structured as follows: Section 2 reviews the literature, Section 3 explains the data and research methodology, while Section 4 shows the empirical findings. Finally, Section 5 summarises the findings and practical implications and identifies the limitations of the paper, sets light to further research and makes recommendations for all stakeholders.

## 2. Literature Review

There are many studies with various findings and results on the impact of sustainability on performance and value of non-financial firms. Eccles et al. [9] suggested that US firms with a higher sustainability measure show better stock market and accounting performance. Esteban-Shances et al. [10] suggested that good corporate governance and good employee relationship are positively associated with firm performance. They base their findings on a sample of 154 financial entities from 22 countries. In the meta-analysis conducted by Albertini [11], a positive relationship was found between corporate environmental and financial performance.

Lima Crisostomo et al. report that corporate social responsibility negatively affects firm value, as measured by Tobin's Q for Brazil firms [12]. Sun et al. reported that there exists an inverted U-shaped relationship between CSR and shareholder value, as measured by Tobin's Q for 468 firms that publicly traded on the US securities exchange [8]. Harjoto and Laksmana concluded that corporate social responsibility has a positive influence on US firm value [13]. Gompers et al. suggested that corporate governance improves firm value [14]. Bayrakdaroglu et al. found evidence that corporate governance enhances EVA (Economic Value Added), MVA (Market Value Added) and CVA (Cash Value Added) in Turkish firms [15]. Jakobs et al. examined the effect of environmental performance on shareholder value [16]. They reported that the market does not react significantly to the aggregated CEI (Corporate Environmental Initiatives) and EAC (Environmental Awards and Certifications) announcements. However, they found significant market reactions for certain CEI and EAC subcategories. The market reacted positively to the announcements of philanthropic gifts for environmental causes and ISO 14001 certifications, but negatively to voluntary emission reductions. Fisher-Vanden and Thorburn found that there was a decrease in the market value of companies joining Climate Leaders and announcing a greenhouse gas reduction goal [17].

Nollet et al. found that there is a U-shaped association between ESG, GPS and financial performance indicators (return on assets and return on capital) [18]. However, they did not find any significant non-linear relationship between EPS, SPS and financial performance measures for S&P 500 firms. Gholami et al. examined the relationship between ESG and profitability for financial and non-financial firms in Australia [4]. They suggested that higher ESG performance is associated with higher firm profitability. However, this relationship differed between financial and non-financial firms. Atan et al. showed that there is no association between ESG and its sub-components and firm value, as measured by Tobin's Q for Malaysian firms [19]. Keceli and Cankaya did not find a significant association between ESG and stock price change in the Nordic and Latin European sample [20].

Examining the studies on the banking industry, Wu and Shen found that corporate social performance positively affected the financial performance of banks in the sample of 162 banks in 22 countries [21]. Siueia et al. reported that CSR positively affected the financial performance in the Sub-Saharan Africa banking industry [22]. Simpson and Kohers found that there was a positive relationship between the bank's social performance and Return on Assets [23]. Soana found that there is no evidence of a significant correlation between CSR and the financial performance in the İtalian banking industry [24].

Carnevale and Mazzuca investigated the relationship between sustainability reports and bank value in the European stock markets [25]. They found a negative impact on stock price and an insignificant impact on earnings per share. Bolton found a positive relation between CSR and bank value, as measured by Tobin's Q in the US banking industry [26]. Peni and Vahamaa found that banks with stronger corporate governance mechanisms had higher profitability in 2008 and lower Tobin's Q and stock return during the crisis (2008–2009) and higher stock returns in the post-crisis period, based on the sample of the large publicly traded US banks [27].

Most previous studies examining the impact of ESG on bank value and performance have assumed that the relationship is linear. Some of these studies reported that the effect of environmental, social and management performance was positive, while others reported that it was negative or neutral. For example, Brogi and Lagasio demonstrated that there is a positive relationship between ESG and ROA in the sample of US banks [28]. Shakil et al. show that EPS and SPS positively affect Return of Equity (ROE), but GPS does not impact the ROE in the sample of emerging market banks [29]. Simsek and Cankaya found that SPS positively impacts ROA and ROE, whereas EPS have a negative effect, and GPS have no impact on the banks in G8 countries [30]. Additionally, very few studies investigate the relationship between ESG and bank value. There are mixed results from previous studies investigating the effect of ESG and ESG sub-components scores on bank value. Di Tommaso and Thornton found that ESG scores and their sub-components are negatively related to bank value measured using Tobin's Q, capital book value, and the stock price of European banks [6]. Miralles-Quiros et al. reported that EPS and GPS performance of banks have a positive impact, but SPS performance has a negative effect on shareholder value creation measured using Tobin's Q in the sample of 166 banks from 31 countries [5]. The finding obtained from regression results of Buallay's study indicated that ESG, EPS and SPS positively affect Tobin's Q, but GPS does not significantly impact the European banking industry [3]. Bually et al. investigate the association between ESG and performance of MENA banks. The results of the regression show that ESG has a negative influence on market value, as measured by Tobin's Q [31]. Similarly to our study, Ahmad et al., used the market value of the companies as a financial performance for FTSE350 UK firms. They reveal that the market value of the firms has a positive and significant effect on ESG, but the results of estimated models for EPS, SPS and GPS are mixed [32].

The results of previous studies show that there is no consensus about the firm's ESG performance on bank value. The relationship between ESG and bank value may be non-linear rather than linear. As far as we know, only one study [7] investigates the non-linear

relationship between the ESG and its sub-components on the bank value. El Khoury found a non-linear relationship between ESG, EPS, SPS and market performance of banks in the MENAT region [7]. This paper reported an inverted U-shaped relationship between ESG, SPS and bank market performance, which is measured by Tobin's Q and stock return. On the other hand, EPS and market performance have a U-shaped relationship, and GPS has no effect on market performance.

## 3. Materials and Methods

The sample employed in this study is an unbalanced panel of 151 US commercial banks from 2016 to 2020. Bank-level financial data are obtained from the Thomson Reuters Eikon database to generate the variables specified in Models (1)–(4). To determine the linkage between ESG and its sub-elements (i.e., EPS, SPS, and GPS) and bank market value, we estimate the following regression models:

$$(MV)_{it} = \alpha_0 + \alpha_1 (MV)_{it-1} + \alpha_2 (ESG)_{it-1} + \alpha_3 (ESG)^2_{it-1} + \alpha_4 (BLVs)_{it-1} \\ + \alpha_5 (COVID-19)_t + \varepsilon_{it} \tag{1}$$

$$(MV)_{it} = \alpha_0 + \alpha_1 (MV)_{it-1} + \alpha_2 (EPS)_{it-1} + \alpha_3 (EPS)^2_{it-1} + \alpha_4 (BLVs)_{it-1} \\ + \alpha_5 (COVID-19)_t + \varepsilon_{it} \tag{2}$$

$$(MV)_{it} = \alpha_0 + \alpha_1 (MV)_{it-1} + \alpha_2 (SPS)_{it-1} + \alpha_3 (SPS)^2_{it-1} + \alpha_4 (BLVs)_{it-1} \\ + \alpha_5 (COVID-19)_t + \varepsilon_{it} \tag{3}$$

$$(MV)_{it} = \alpha_0 + \alpha_1 (MV)_{it-1} + \alpha_2 (GPS)_{it-1} + \alpha_3 (GPS)^2_{it-1} + \alpha_4 (BLVs)_{it-1} \\ + \alpha_5 (COVID-19)_t + \varepsilon_{it} \tag{4}$$

In Models (1)–(4), the dependent variable is $(MV)_{it}$ and this variable is measured by the logarithm of the market value. $\alpha_0$ is the intercept. $(MV)_{it-1}$ is the one-year lagged dependent variable. $(ESG)_{it}$ and $(ESG)^2_{it}$ are ESG and the squared term of ESG scores. Likewise, $(EPS)_{it}$ and $(EPS)^2_{it}$ are EPS scores and the squared term of EPS scores. $(SPS)_{it}$ and $(SPS)^2_{it}$ are SPS scores and the squared term of SPS scores, and $(GPS)_{it}$ and $(GPS)^2_{it}$ are GPS scores and the squared term of GPS scores. $(BLVs)_{it}$ represents bank-level control variables: size, beta, capital adequacy ratio, return on assets, income diversity, non-performing loans ratio, and risk-weighted assets ratio. $(COVID-19)_t$ is a dummy variable representing the COVID-19 pandemic crisis. $\varepsilon_{it} = f_i + \omega_t + u_{it}$, $f_i$ denotes the bank fixed effect, $\omega_t$ is the time-fixed effect, and $u_{it}$ is the idiosyncratic error term. Definitions of all variables used in the study are given in Table 1.

We estimate quadratic regression Models (1)–(4) employing a fixed-effects regression according to Hausman test statistics. Moreover, standard errors at the bank level are clustered based on the results of heteroskedasticity (Modified Wald) and autocorrelation (Wooldridge test) tests. Moreover, all explanatory variables included in the quadratic regression models are lagged one year, except for COVID-19 dummy variable to mitigate the potential endogeneity issue.

Two basic performance measures are accounting-based and market-based. Accounting-based performance measures such as ROA and ROE are affected by accounting procedures and accounts manipulations. For investors, not only accounting-based performance measures but also the market-based performance measures of companies are extremely important. For this reason, market-based performance measures such as Tobin's Q, price to book value, and stock return are frequently used to determine firm performance. Previous studies investigating the effect of the ESG on bank value generally use Tobin's Q, price to book value and stock return. The stock price or market value of firms is one of the most important determinants of the investment decisions for stock market investors. Unlike previous studies, the market value variable was used to represent bank value in this study.

**Table 1.** Variable definitions.

| Variable | Notation | Definition |
|---|---|---|
| **Dependent Variable** | | |
| Market Value | MVB | Market value is the share price multiplied by the number of ordinary shares in the issue. The amount in issue is updated whenever new tranches of stock are issued or after a capital change. |
| **Independent Variables** | | |
| ESG Score | ESG | ESG Score is an overall company score based on the self-reported information in the environmental, social and corporate governance pillars. |
| EPS Score | EPS | EPS is a company's weighted average relative rating based on the reported environmental information and the resulting three environmental category scores. |
| SPS Score | SPS | SPS is the weighted average relative rating of a company based on the reported social information and the resulting four social category scores. |
| GPS Score | GPS | GPS is a company's weighted average relative rating based on the reported governance information and the resulting three governance category scores. |
| **Bank-Level Control Variables** | | |
| Employees | Size | Employees represent the number of both full and part-time employees of the company. |
| Beta | Beta | Historical beta |
| Capital Adequacy Ratio | CAR | The capital adequacy ratio represents the ratio of total capital to total risk-weighted assets, calculated in accordance with banking regulations and expressed as a percentage. |
| Return on Assets Ratio | ROA | Net income to total assets |
| Income Diversity | INCDIV | Non-Interest Income/Total Operating IncomeTotal Operating Income = Net Interest Income + Non-Interest Income |
| Non-Performing Loans Ratio | NPL | Non-Performing Loans/Total Loans |
| Risk-Weighted Assets Ratio | RWA | Risk-weighted assets represent the total carrying value of each asset class multiplied by their assigned risk weighting, as defined by banking regulations. This item may also be referred to as risk-adjusted assets. |
| **Crisis Control Variable** | | |
| COVID-19 | COVID-19 | Dummy variable for the year 2020 |

Source: Authors' Compilation.

## 4. Empirical Results

Table 2 reports descriptive statistics of all variables employed for our study. The dependent variable of the study is MV. We transform this variable into a natural logarithmic form. The mean of the variable MV is about 7. The averages of the ESG variable and its sub-components are approximately 35, 22, 50, and 33, respectively. The significant differences observed between the maximum and minimum values of both dependent and independent variables reveal that there are serious differences between the banks in the sample in terms of these variables.

The independent and control variables' multi-collinearity issue is checked by employing Pearson's correlation. Table 3 demonstrates the results of the correlation analysis. When the correlation coefficients calculated between the independent variables are examined, high correlations are observed between the ESG and GPS and SPS variables. In line with this result, it is decided to model the ESG variable and its sub-elements separately to minimise the multi-collinearity issue. Considering the correlation coefficients calculated between the control variables, it is determined that the highest correlation coefficient calculated between these variables is 0.693. This result displays that multi-collinearity will not be a serious problem in terms of control variables for multivariate analysis.

The estimation results of linear and non-linear models are reported in Table 4. Columns 1, 3, 5 and 7 present linear relationships between ESG and its sub-elements and bank market value. In addition, columns 2, 4, 6 and 8 indicate the non-linear associations between ESG and its sub-components and the bank market value. The results of linear models show no

statistically significant relationship between the ESG, EPS, GPS and SPS variables and the bank's market value.

**Table 2.** Descriptive statistics.

| Variable | Mean | Median | SD | Min. | Max. | N |
|---|---|---|---|---|---|---|
| Ln (MV) | 6.8758 | 6.7010 | 1.6621 | 3.1822 | 12.9119 | 879 |
| ESG | 35.3028 | 32.94 | 13.8379 | 5.31 | 88.45 | 830 |
| EPS | 22.1143 | 21.14 | 23.0366 | 0.52 | 94.99 | 372 |
| GPS | 49.6633 | 51.295 | 18.9178 | 3.85 | 92.09 | 830 |
| SPS | 33.1854 | 30.63 | 15.9685 | 1.27 | 90.96 | 830 |
| Ln (SIZE) | 7.0608 | 6.7581 | 1.4977 | 4.4544 | 12.5028 | 837 |
| BETA | 1.0475 | 1.0728 | 0.4823 | −0.1729 | 2.5315 | 877 |
| CAR | 14.4299 | 14.015 | 2.2481 | 8.0 | 26.57 | 868 |
| ROA | 1.1907 | 1.18 | 0.4131 | −4.32 | 2.26 | 801 |
| NPL | 1.1644 | 0.8176 | 2.0487 | 0.0093 | 53.8776 | 863 |
| INCDIV | 23.3548 | 23.1818 | 9.8184 | 1.7603 | 55.9979 | 880 |
| RWA | 0.7301 | 0.7583 | 0.1632 | 0 | 1.2903 | 880 |

Source: Authors' Compilation.

**Table 3.** Correlation matrix.

| | ESG | EPS | GPS | SPS | SIZE | BETA | CAR | ROA | NPL | INCDIV | RWA |
|---|---|---|---|---|---|---|---|---|---|---|---|
| ESG | 1.000 | | | | | | | | | | |
| EPS | 0.190 * | 1.000 | | | | | | | | | |
| GPS | 0.771 * | 0.097 | 1.000 | | | | | | | | |
| SPS | 0.746 * | 0.183 * | 0.212 * | 1.000 | | | | | | | |
| SIZE | 0.421 * | 0.343 * | 0.157 * | 0.455 * | 1.000 | | | | | | |
| BETA | 0.437 * | 0.283 * | 0.215 * | 0.424 * | 0.693 * | 1.000 | | | | | |
| CAR | −0.137 | −0.234 * | −0.114 | −0.132 | −0.137 | −0.088 | 1.000 | | | | |
| ROA | 0.076 | −0.115 | 0.044 | 0.050 | 0.045 | 0.038 | 0.089 | 1.000 | | | |
| NPL | −0.006 | −0.032 | −0.042 | 0.039 | 0.080 | 0.021 | 0.062 | −0.266 * | 1.000 | | |
| INCDIV | 0.034 | 0.115 | −0.043 | 0.115 | 0.356 * | 0.068 | −0.035 | −0.121 | 0.246 * | 1.000 | |
| RWA | −0.110 | −0.000 | −0.049 | −0.135 | −0.031 | −0.009 | 0.027 | −0.137 | −0.022 | −0.020 | 1.000 |

Source: Authors' Compilation. * $p < 0.10$.

**Table 4.** Estimation results of linear and non-linear models.

| | (1) | (2) | (3) | (4) | (5) | (6) | (7) | (8) |
|---|---|---|---|---|---|---|---|---|
| ESG | −0.00133 (0.00179) | 0.0134 ** (0.0056) | | | | | | |
| ESG² | | −0.0002 ** (0.00009) | | | | | | |
| EPS | | | 0.00006 (0.00331) | −0.0094 * (0.0053) | | | | |
| EPS² | | | | 0.00018 ** (0.00008) | | | | |
| GPS | | | | | 0.00001 (0.0007) | 0.0006 (0.0029) | | |
| GPS² | | | | | | −0.000006 (0.00003) | | |
| SPS | | | | | | | −0.0008 (0.0017) | 0.0071 ** (0.0034) |
| SPS² | | | | | | | | −0.0001 ** (0.00005) |
| MV$_{t−1}$ | 0.235 *** (0.0608) | 0.229 *** (0.0601) | 0.403 *** (0.120) | 0.403 *** (0.113) | 0.238 *** (0.0611) | 0.237 *** (0.0617) | 0.239 *** (0.0608) | 0.233 *** (0.0614) |
| SIZE | 0.286 *** (0.0963) | 0.291 *** (0.0940) | 0.374 *** (0.123) | 0.301 ** (0.125) | 0.286 *** (0.0974) | 0.288 *** (0.0964) | 0.290 *** (0.0982) | 0.300 *** (0.0978) |

**Table 4.** *Cont.*

|  | (1) | (2) | (3) | (4) | (5) | (6) | (7) | (8) |
|---|---|---|---|---|---|---|---|---|
| BETA | −0.0487 (0.0616) | −0.0425 (0.0620) | −0.241 ** (0.115) | −0.222 ** (0.102) | −0.0517 (0.0616) | −0.0509 (0.0616) | −0.0517 (0.0623) | −0.0474 (0.0626) |
| CAR | 0.0343 ** (0.0163) | 0.0330 ** (0.0162) | 0.0462 (0.0311) | 0.0386 (0.0270) | 0.0348 ** (0.0162) | 0.0347 ** (0.0163) | 0.0345 ** (0.0164) | 0.0350 ** (0.0164) |
| ROA | 0.119 ** (0.0592) | 0.127 ** (0.0597) | −0.00294 (0.0989) | 0.0432 (0.0869) | 0.117 ** (0.0585) | 0.117 ** (0.0582) | 0.117 ** (0.0589) | 0.122 ** (0.0595) |
| INCDIV | −0.00130 (0.00371) | −0.00161 (0.00369) | 0.0189 *** (0.00669) | 0.0112 * (0.00622) | −0.00120 (0.00371) | −0.00121 (0.00371) | −0.00130 (0.00374) | −0.00203 (0.00373) |
| NPL | 0.0106 (0.0272) | 0.0160 (0.0264) | −0.0488 (0.0345) | −0.0605 * (0.0317) | 0.0116 (0.0267) | 0.0117 (0.0267) | 0.0109 (0.0274) | 0.0121 (0.0272) |
| RWA | 0.0693 (0.114) | 0.0636 (0.105) | −0.235 (0.788) | −0.0966 (0.770) | 0.0725 (0.110) | 0.0743 (0.109) | 0.0689 (0.112) | 0.0436 (0.115) |
| COVID-19 | −0.460 *** (0.0349) | −0.459 *** (0.0349) | −0.534 *** (0.102) | −0.554 *** (0.103) | −0.463 *** (0.0346) | −0.463 *** (0.0345) | −0.462 *** (0.0342) | −0.466 *** (0.0342) |
| Intercept | 2.888 *** (0.583) | 2.681 *** (0.569) | 1.289 (0.968) | 2.009 ** (0.939) | 2.813 *** (0.590) | 2.799 *** (0.577) | 2.817 *** (0.566) | 2.700 *** (0.565) |
| Bank fixed effects | Yes | Yes | Yes | Yes | Yes | Yes | Yes | Yes |
| Time fixed effects | Yes | Yes | Yes | Yes | Yes | Yes | Yes | Yes |
| Hausman test | 117.27 *** | 119.11 *** | 50.53 *** | 51.61 *** | 112.14 *** | 111.99 *** | 117.71 *** | 117.84 *** |
| Modified Wald test | $3.9 \times 10^{30}$ *** | $6.1 \times 10^{29}$ *** | $8.0 \times 10^{33}$ *** | $3.1 \times 10^{32}$ *** | $1.8 \times 10^{30}$ *** | $1.5 \times 10^{30}$ *** | $2.8 \times 10^{5}$ *** | $1.9 \times 10^{5}$ *** |
| Wooldridge test | 61.532 *** | 59.525 *** | 18.518 *** | 20.342 *** | 61.460 *** | 59.933 *** | 60.989 *** | 60.870 *** |
| Within R-Squared | 0.6365 | 0.6422 | 0.7274 | 0.7581 | 0.6358 | 0.6358 | 0.6361 | 0.6400 |
| F-test | 46.88 *** | 46.05 *** | 21.70 *** | 26.47 *** | 47.87 *** | 45.74 *** | 47.67 *** | 49.60 *** |
| N | 515 | 515 | 196 | 196 | 515 | 515 | 515 | 515 |
| Banks | 151 | 151 | 112 | 112 | 151 | 151 | 151 | 151 |

Note: Columns (1), (3), (5), (7) show the results of linear models and, columns (2), (4), (6), (8) show the results of non-linear models for ESG, EPS, SPS and GPS, respectively. All explanatory variables are lagged by one year, except for COVID-19 dummy variable. Robust standard errors are reported in the parentheses. * $p < 0.10$, ** $p < 0.05$, *** $p < 0.01$. Source: Authors' Compilation.

However, the estimation results of the quadratic models imply that there is a non-linear relationship between ESG and ESG sub-components and bank market value, except for the GPS. More specifically, Consistent with El Khoury et al., an inverted U-shaped non-linear association is found between the ESG and SPS variables and the bank market value. On the other hand, it is determined that there exists a U-shaped non-linear relationship between the EPS variable and the bank's market value [7].

The inverted U-shaped association between ESG and bank market value implies that the market value enhances with an increase in ESG investments. However, bank market value begins to decrease with an increase in ESG investments after a certain level. In other words, ESG investments have a positive impact in the short run, but this effect turns negative in the long run. The inverted U-shaped association confirms the findings of El Khoury et al. It can be recommended that banks should determine turning points where the impact of ESG investments on market value turns negative to rationalise the ESG investments [7] and shareholder value creation. Attracting more environmentally-conscious consumers by using green marketing strategies and increasing demand for stocks due to the continuous increase in the number of investors who care about sustainability may increase the bank's value in the long run.

The results of the U-shaped relationship investigated between EPS and bank market value suggest that increasing environmental investments up to a threshold level negatively influences bank market value. Investments made in the beginning may lead to a decrease in the market value of the banks owing to the increase in costs and the fact that the bank's environmentally-conscious investments are not highlighted enough. However, continuing environmental investments beyond the threshold level enhance bank market value. In

short, the negative effect in the short run turns positive in the long run. Thus, it can be said that environmental investments positively affect the bank market value in the long run. In order for environmental investments to benefit stakeholders, managers can be advised to allocate more resources to environmental investments within the framework of long-term strategic plans.

The inverted U-shaped correlation between SPS and bank market value means that increasing socially responsible investments up to a certain point contributes to maximising the shareholder value, but continuing this investment after this point negatively affects the bank value. It can be said that after a certain point, the costs of socially responsible investments exceed the benefits. Developing policies that will make socially responsible projects more visible by managers may increase the demand of socially responsible customers and investors for bank services and stocks. Thus, the negative impact of socially responsible investments on bank value in the short run can be reduced, and at the same time, its positive impact can be achieved earlier.

We found that there was no statistically significant influence of the GPS variable on the market value. Buallay and El Khoury et al. reported similar results in their study [3,7]. However, disclosing more information about the activities carried out to improve corporate governance can positively affect the bank's performance and increase its market value [3].

Developing policies that encourage and support the companies and increase the awareness of both investors and individuals in the environmental, social and management fields can reduce the negative effects and increase the positive effects on the market value of the ESG and its sub-elements.

The finding of the control variables suggested that bank managers should increase the capital adequacy ratio, profitability and income diversity while avoiding non-performing loans to maximise bank market value. The bank's capital position may be value-relevant because of several factors [33]: Firstly, with high capital ratios, bank holding companies pay lower FDIC insurance costs, incur lower regulatory costs and risks, and have higher elasticity in activities and greater capability to grow. Second, high capital ratios can accumulate capital to facilitate value-creating growth. Third, excess capital reflects the firm's market power because banks with large market power believe that they will suffer more from regulatory interventions compared to other banks. As a result, they have a greater motivation to retain excess capital. These effects indicate that the market-to-book ratio should be positively correlated with capital adequacy. All models except Models (3) and (4) indicated that the capital adequacy ratio has a positive and statistically significant influence on bank market value. This means that the higher capitalised banks tend to have higher market value. The results support the findings of Miralles et.al. and Di Tommaso and Thornton, [5,6].

Larger banks may have more market power, benefit from the diversification of revenues and easily access the capital market for the capital requirements. However, the impact of size can turn out to be adverse for extremely large banks due to diseconomies of scale. For this reason, the effect of size on market value is ambiguous [5,6,34–38]. Former studies [5,6,34,35] revealed that bank size affects bank value negatively. Moreover, Avramidis et al. [38] found an inverted U-shaped relationship between size and market performance. However, Kuzucu and Kuzucu could not find any significant relationship between size and bank value [37]. Besides, our paper's findings comply with Tui et al. [36]. The impact of size on bank market value is positive and statistically significant for all models. This result stated that an increase in the size of banks enhances the market value.

The increase in banks' profitability is expected to affect the market value positively. Previous literature has shown that firm value is directly related to firm performance [5,6,34,36,37,39–41]. The results of this study support the previous literature. All models except Models (3) and (4) show that return on assets has a positive and statistically significant impact on bank market value.

The bank's income sources can be divided into interest and non-interest incomes. Interest income includes lending activity such as interest earned from loans and investment securities. Non-interest income covers non-lending activities such as investment banking,

asset management and insurance underwriting, fee-paying and commission-paying services, trading and derivatives and other non-interest income activities [2]. Income diversity is the banks' diversification degree between lending and non-lending activities. Banks must diversify their sources of net operating income between net interest income and non-interest income. A bank is fully diversified when there is a balance between the two types of income [7]. Portfolio diversification would increase the opportunities for overlapping assets, decreasing individual risks and increasing systematic risks. Decreasing individual risks raises the bank values, but increasing systematic risks reduces bank value.

For this reason, full diversification is not always optimal [42]. Although several studies have investigated the influence of income diversity on bank value, they have found contradictory results. For example, Sawada and Minton et al. find that non-interest income positively affects bank value [35,43]. Tsai et al. reported that more diversification leads to lower bank value [42]. Whereas, Yildirim and Efthyvoulou have not found a significant relationship [44]. The coefficient of income diversity variable is positive and significant only in Models (3) and (4). It can be said that increases in income diversity, measured by non-interest income to total operating income, enhance the market value of banks.

The non-performing loans to total loans ratio is a measure of the quality of loans. Therefore, non-performing loans to total loan ratios are expected to affect the bank value adversely. However, no statistically negative and significant relationship was found in models except Model (3). However, it can be said that this result provides evidence that non-performing loans have a decreasing effect on bank market value, in line with [6,40,44]. Furthermore, Sawada also found that the impact of non-performing loans on market value is mostly not statistically significant, similarly to this study [43].

The coefficients on the one lag of bank market value variable for all models are positive and statistically significant, in line with [6]. The results suggest that the one lag of bank market value positively impacts current market value. Our analyses reveal that the beta coefficient has a negative and statistically significant effect on the bank market value for only Models (3) and (4). Following the study by Kuzucu and Kuzucu, we use risk-weighted assets to total assets ratio to measure banks' risk-taking behaviour [37]. However, we have not found a statistically significant relationship between bank market value and risk-weighted assets to total assets ratio.

The sample period of this study also covers the COVID-19 pandemic period. The COVID-19 pandemic was included in the study as a dummy variable as it may impact firm value. The results of all models show that the COVID-19 pandemic negatively and significantly influences bank market value. The result suggests that the COVID-19 pandemic caused a decrease in bank market value.

In order to check the robustness of our results, we employ an alternative estimator called Driscoll-Kraay standard errors estimator using fixed effects regression taking into account autocorrelation, heteroscedasticity, and cross-section dependence [45]. The results from the equations we re-estimated using the Driscoll-Kraay standard errors estimator are presented in the Appendix A. After evaluating these results it was determined that the estimation results obtained from the Driscoll-Kraay standard errors estimator for the ESG and its sub-components are similar to those of the Fixed Effects estimator [46].

## 5. Conclusions

Very few studies investigate the effect of ESG performance on bank value, and most of these studies relate a linear relationship between ESG and bank value. To the best of our knowledge, no study uses banks' market value as a measure of bank value. We have tried to close this gap in the literature by examining the impact of ESG and ESG sub-components on U.S. commercial bank market value by using linear and non-linear models. Additionally, the effect of COVID-19 was taken into account in the study. Our sample includes 176 US commercial banks from 2016 to 2020.

No significant linear relationship has been found in this study. Empirical results of non-linear models suggested that there is a statistically significant inverted U-shape

relationship between ESG and SPS scores and bank market value. However, the results obtained from non-linear models for EPS and GPS have changed. There exists a U-shaped relationship between EPS and bank market value. At the same time, GPS has no statistically significant effect on bank market value. Additionally, the findings for the control variables evidence that capital adequacy ratio, profitability, income diversity and size have a positive impact, and non-performing loans, beta coefficient and the COVID-19 pandemic have a negative impact on bank market value.

There are bank-specific and external factors that affect the market value of banks. This study contributes to a better understanding of the effect of ESG performance on bank market value for investors, managers, regulators and other stakeholders. Developing policies that will make ESG investments more visible and known, such as highlighting them in advertisements, may contribute to increasing the positive effect of ESG on bank value. This is because individuals with environmental and social sensitivity will prefer these banks to purchase services and invest in stocks. It is recommended that the policy makers and regulators provide more support to increase the awareness of all stakeholders and encourage the companies in the environmental, social and management fields. Empirical findings provide evidence that sustainability activities can increase bank value. In addition, it is known that sustainability activities are very important for the survival of the firms and the protection of the ecosystem, and help to improve the social justice and the sustainable economic growth of the countries. For this reason, it is recommended that policy makers develop policies that are more encouraging in the manner that they regulate sustainability activities of firms.

The main limitation of this paper is that it covers only U.S. commercial banks. Therefore, it is suggested that future studies can investigate the influence of ESG on bank market value in other continents or regions. The effect of moderating variables such as ownership structure in the relationship between ESG and bank value can also be examined. One can investigate whether the effect of ESG on firm value differs by sector.

**Author Contributions:** Conceptualization, E.E. and E.Ö.; methodology, E.E., S.G. and E.Ö.; software, E.E., E.Ö., S.G. and B.S.; validation, S.G. and B.S.; formal analysis, E.E. and E.Ö.; investigation, E.E., S.G. and B.S.; resources, S.G. and B.S.; data curation, E.E.; writing—original draft preparation, E.E., E.Ö., S.G., I.R. and B.S.; writing—review and editing, S.G., I.R. and B.S.; visualisation, S.G., B.S. and I.R. supervision, E.E., E.Ö., S.G. and B.S. All authors have read and agreed to the published version of the manuscript.

**Funding:** This research received no external funding.

**Institutional Review Board Statement:** Not applicable.

**Informed Consent Statement:** Not applicable.

**Data Availability Statement:** Data found in the paper itself are available from the correspondent author.

**Conflicts of Interest:** The authors declare no conflict of interest.

## Appendix A

**Table A1.** Estimation results of Driscoll-Kraay Standart Errors Estimator.

|  | (1) | (2) | (3) | (4) | (5) | (6) | (7) | (8) |
|---|---|---|---|---|---|---|---|---|
| ESG | 0.000163 (0.000982) | 0.0134 ** (0.00368) |  |  |  |  |  |  |
| ESG$^2$ |  | −0.000221 ** (0.0000495) |  |  |  |  |  |  |
| EPS |  |  | −0.000401 (0.00239) | −0.00965 *** (0.00164) |  |  |  |  |
| EPS$^2$ |  |  |  | 0.000179 *** (0.0000149) |  |  |  |  |

**Table A1.** *Cont.*

|  | (1) | (2) | (3) | (4) | (5) | (6) | (7) | (8) |
|---|---|---|---|---|---|---|---|---|
| GPS |  |  |  |  | 0.0000131 (0.000284) | 0.000555 (0.00205) |  |  |
| GPS$^2$ |  |  |  |  |  | −0.00000583 (0.0000217) |  |  |
| SPS |  |  |  |  |  |  | −0.000788 (0.000637) | 0.00711 * (0.00230) |
| SPS$^2$ |  |  |  |  |  |  |  | −0.000122 ** (0.0000242) |
| MV$_{t-1}$ | 0.235 * (0.0990) | 0.229 (0.0980) | 0.886 *** (0.0265) | 0.883 ** (0.0296) | 0.238 * (0.101) | 0.237 * (0.100) | 0.239 * (0.0991) | 0.233 * (0.0972) |
| SIZE | 0.286 *** (0.0789) | 0.291 *** (0.0751) | 0.374 *** (0.0403) | 0.301 ** (0.0436) | 0.286 *** (0.0762) | 0.288 *** (0.0737) | 0.290 *** (0.0783) | 0.300 *** (0.0734) |
| BETA | −0.0487 ** (0.0117) | −0.0425** (0.0129) | 0.00337 (0.0376) | −0.000389 (0.0367) | −0.0517 ** (0.0117) | −0.0509 ** (0.0129) | −0.0517 ** (0.0121) | −0.0474 ** (0.0141) |
| CAR | 0.0343 ** (0.00785) | 0.0330 ** (0.00770) | 0.0122 (0.0207) | 0.0118 (0.0207) | 0.0348 ** (0.00807) | 0.0347 ** (0.00814) | 0.0345 ** (0.00769) | 0.0350 ** (0.00825) |
| ROA | 0.119 ** (0.0310) | 0.127 ** (0.0264) | −0.0159 (0.0324) | −0.00207 (0.0340) | 0.117 ** (0.0299) | 0.117 ** (0.0277) | 0.117 ** (0.0306) | 0.122 ** (0.0293) |
| INCDIV | −0.00130 (0.00123) | −0.00161 (0.00124) | −0.000760 (0.00144) | −0.00127 (0.00138) | −0.00120 (0.00119) | −0.00121 (0.00118) | −0.00130 (0.00123) | −0.00203 (0.00122) |
| NPL | 0.0106 (0.0113) | 0.0160 (0.0107) | −0.0197 (0.0263) | −0.0215 (0.0259) | 0.0116 (0.0113) | 0.0117 (0.0112) | 0.0109 (0.0109) | 0.0121 (0.0111) |
| RWA | 0.0693 (0.0398) | 0.0636 (0.0356) | −0.223 *** (0.0323) | −0.228 *** (0.0349) | 0.0725 (0.0396) | 0.0743 (0.0327) | 0.0689 (0.0381) | 0.0436 (0.0462) |
| COVID-19 | −0.460 *** (0.0144) | −0.459 *** (0.0143) | −0.572 *** (0.0106) | −0.580 *** (0.00378) | −0.463 *** (0.0146) | −0.463 *** (0.0142) | −0.462 *** (0.0135) | −0.466 *** (0.0136) |
| Intercept | 2.888 *** (0.330) | 2.681 *** (0.367) | 0.628 (0.269) | 0.641 (0.301) | 2.813 *** (0.368) | 2.799 *** (0.356) | 2.817 *** (0.330) | 2.700 *** (0.370) |
| Bank fixed effects | Yes | Yes | Yes | Yes | Yes | Yes | Yes | Yes |
| Time fixed effects | Yes | Yes | Yes | Yes | Yes | Yes | Yes | Yes |
| Within R-Squared | 0.6365 | 0.6422 | 0.7274 | 0.7581 | 0.6358 | 0.6358 | 0.6361 | 0.6400 |
| F-test | 10.46 ** | 7.53 * | 16.50 ** | 2.00 | 47.87 *** | 45.74 *** | 47.67 *** | 49.60 *** |
| N | 515 | 515 | 196 | 196 | 515 | 515 | 515 | 515 |
| Banks | 151 | 151 | 112 | 112 | 151 | 151 | 151 | 151 |

Note: Columns (1), (3), (5), (7) show the results of linear models and, columns (2), (4), (6), (8) show the results of non-linear models for ESG, EPS, SPS and GPS, respectively. All explanatory variables are lagged by one year, except for COVID-19 dummy variable. Robust standard errors are reported in the parentheses. * $p < 0.10$, ** $p < 0.05$, *** $p < 0.01$. Source: Authors' Compilation.

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
