# Peer review of "The Impact of ESG Scores on Bank Market Value? Evidence from the U.S. Banking Industry"

_sustainability, doi:10.3390/su14159527_

Round 1

Reviewer 1 Report

1. The way references are referred to in main text is not standardized and should be standardized.

- For example, there are no "Friedman et al.(2005)"Freeman(1984)" at the reference of last part

- The order of [4][6][5] is wrong.

- No name before [6] [8], but there are name before [5] [9]

2.

Rate of Assets (line 124) or Return on Assets Ratio (P5 table)

Which is right?

Please review all throughout the text.

3.

ROE: Return of Equity (line90) and Rate of Equity(line125). Which is right?

Please review all throughout the text.

4. (line 128-134)

No Forthly.? Thirdly, next Fifth?

5. Table4

Robustness checks should be provided

6. What do SPS and EPS in abstract stand for?

 Sales Per Share (SPS) and Earnings per 21 Share (EPS)? Really?

What is the difference between SPS and EPS of line 34?

7. Overall, the writing style is quite crude. There is no uniformity, and the style is not consistent. You should review the entire text.

Author Response

Thank you for the very valid comments and suggestions, which we have agreed to and incorporated in our paper. These suggestions and comments have made our paper flow much better and gave it more stability and strength in its value.  For ease of the reader we have written our answers to your comments and suggestions below  as authors’ response.

  1. The way references are referred to in main text is not standardized and should be standardized.

- For example, there are no "Friedman et al.(2005)"Freeman(1984)" at the reference of last part

- The order of [4][6][5] is wrong.

- No name before [6] [8], but there are name before [5] [9]

Authors’ response: Agreed. We made all the adjustments as requested

  1. Rate of Assets (line 124) or Return on Assets Ratio (P5 table)

Which is right?

Please review all throughout the text.

Authors’ response: Agreed. We have adjusted to Return on Assets and Return on Equity

3.ROE: Return of Equity (line90) and Rate of Equity(line125). Which is right?

Please review all throughout the text.

 Authors’ response: Agreed. We have adjusted to Return on Assets and Return on Equity

  1. (line 128-134)

No Forthly.? Thirdly, next Fifth?

 Authors’ response: Agreed. This paragraph has been reviewed.

  1. Table 4 – Robustness checks should be provided

Authors’ response: Agreed. We have provided the robustness checks by applying Driscoll-Kraay Standard Errors Estimator and reported them in the appendix. We have also discussed these in the text vide track changes.

  1. What do SPS and EPS in abstract stand for?

 Sales Per Share (SPS) and Earnings per 21 Share (EPS)? Really?

Authors’ response: Agreed. Was due to an autocorrect - SPS is Social Pillar Score and EPS is Environment Pillar Score

What is the difference between SPS and EPS of line 34?  Authors’ response: Agreed. This was explained in the introduction 2nd paragraph.

  1. Overall, the writing style is quite crude. There is no uniformity, and the style is not consistent. You should review the entire text.

 Authors’ response: Agreed. The entire text has been reviewed and edited by a native English speaker for clarity, grammar, and flow.

Reviewer 2 Report

After review of the manuscript “The Impact of ESG Scores on Bank Market Value? Evidence from the U.S. Banking Industry” I personally like to appreciate your efforts to present your research work. But before your work will be recommended, some comments must be incorporated to improve the quality of your work as well as for further publication in a sustainability journal.  

Introduction

  1. A literature review section should be created.
  2. The introduction part is required to add a few more sentences to increase the strength of this article and kindly bring in the research problem, objective, and novelty.

Materials and Methods

The authors used only five years dataset, which is the weakness of the study. I strongly suggest expanding the dataset of the study. Even in their model, they used one lag which reduces the year. This issue must be solved.

Empirical Results Section

The authors clearly report the empirical findings and provide a clear explanation but based on the empirical findings, what do you suggest to the policymakers to do?

Conclusions section

Since this manuscript is submitted to a Sustainability journal, the authors should support their findings based on the sustainability framework while suggesting policies. 

Author Response

Thank you for the very valid comments and suggestions, which we have agreed to and incorporated in our paper. These suggestions and comments have made our paper flow much better and gave it more stability and strength in its value.  For ease of the reader we have written our answers to your comments and suggestions below  as authors’ response.

After review of the manuscript “The Impact of ESG Scores on Bank Market Value? Evidence from the U.S. Banking Industry” I personally like to appreciate your efforts to present your research work. But before your work will be recommended, some comments must be incorporated to improve the quality of your work as well as for further publication in a sustainability journal.  

Introduction

  1. A literature review section should be created.

Authors’ response: Agreed. A Literature section created in line 131 to 217

  1. The introduction part is required to add a few more sentences to increase the strength of this article and kindly bring in the research problem, objective, and novelty.

Authors’ response: Agreed. We have added some sentences to, and reshuffled the introduction, to highlight the gap, objective and the new insights to the sustainability literature. Hopefully we have shown the significance and need of our paper (Vide track changes)

Materials and Methods

The authors used only five years dataset, which is the weakness of the study. I strongly suggest expanding the dataset of the study. Even in their model, they used one lag which reduces the year. This issue must be solved.

Authors’ response: Our choice of this period was determined by the availability of the largest data set of Banks on ESG and its sub-elements of US commercial banks obtained from the Thomson Reuters Eikon database, which is the largest database on the subject. This dataset covers the years 2016-2020.

Moreover, we also chose this period since if the chosen sample period was longer, the number of banks would have been less significant since.

Also, all explanatory variables included in the quadratic regression models are lagged one year except for Covid-19 dummy variable to mitigate the potential endogeneity issue.

Empirical Results Section

The authors clearly report the empirical findings and provide a clear explanation but based on the empirical findings, what do you suggest to the policymakers to do?

Authors’ response:We have added the following paragraph above Table 4 (line 337 to 340).

“Developing policies that encourage and support companies and increase the awareness of both investors and individuals in the environmental, social and management fields can reduce the negative effects and increase the positive effects on the market value of the ESG and its sub-elements”.

Conclusions section

Since this manuscript is submitted to a Sustainability journal, the authors should support their findings based on the sustainability framework while suggesting policies. 

Authors’ response:We have addressed this in the third paragraph of the conclusion which was revised as follows (line 433 to 447);

“There are bank-specific and external factors that affect the market value of banks. This study contributes to a better understanding of the effect of ESG performance on bank market value for investors, managers, regulators and other stakeholders. Developing policies that will make ESG investments more visible and known, such as highlighting them in advertisements, may contribute to increasing the positive effect of ESG on bank value. This is because individuals with environmental and social sensitivity will prefer these banks to purchase services and invest in stocks. It is recommended that the policy makers and regulators provide more support to increase the awareness of all stakeholders and encourage the companies in the environmental, social and management fields. Empirical findings provide evidence that sustainability activities can increase bank value. In addition, it is known that sustainability activities are very important for the survival of the firms, the protection of the ecosystem, help to improve the social justice and the sustainable economic growth of the countries. For this reason, it is recommended that policy makers develop policies that are more encouraging in the manner that they regulate sustainability activities of firms”.

Reviewer 3 Report

1. Vary data collection approach and analysis for comparability.

2. Enhance your citations.

Author Response

Thank you for the very valid comments and suggestions, which we have agreed to and incorporated in our paper. These suggestions and comments have made our paper flow much better and gave it more stability and strength in its value.  For ease of the reader we have written our answers to your comments and suggestions below  as authors’ response.

OBSERVATION AND COMMENT 1

The research focus is on corporate social responsibility practice of US banks to the stakeholders which include:

  • EPS: Host community environmental sustainability (immediate vicinity preservation);
  • SPS: Employees welfare (investment in human resources/capital);
  • GPS: Corporate governance structure and

The study made use of Environmental, Social and Governance (ESG) score elements. I do not have issues with this scoring as a source of data. However, since the researchers have mentioned that previous authors applied the ESG in their studies, l would suggest that a new phase be introduced. That is, the researchers can employ usage of raw financial data from the banks under study, by extracting facts from their annual financial reports. Then panel data regression may be more suitable. Alternatively, the authors can also try out this method to see if they can have the same research outcome by way of comparison.

Authors’ response: As far as we know, this is the first study that examines the impacts of ESG performance on the bank market value. We are not aware of any previous similar study.

OBSERVATION AND COMMENT 2

In the introductory section page 1 line 33-41, the authors made claims that require some scholarly backing. I am of the opinion that there should be citations to substantiate the statements and claims on the 3 components of ESG scores.

The mentioned paragraph has been rewritten and the source has been cited.

OBSERVATION AND COMMENT 3

The reference list is not too appropriate. A well-researched article should include references ranging from 35 and above. I think the authors can improve on the reference section by providing detailed citations.

Authors' response. Agreed, the Reference list has been enhanced further.

Round 2

Reviewer 1 Report

I have looked at the Abstract, and even after a short review, it appears that the points raised have not yet been corrected.

- For example, a first-time abbreviation should indicate what it stands for (what is SPS, EPS?).

-Also, there is a mix of references with and without names in front of the numbers. It should be unified.

Author Response

Thank you for the very valid comments and suggestions, which we have agreed to and incorporated in our paper. These suggestions and comments have made our paper flow much better and gave it more stability and strength in its value.  For ease of the reader, we have written our answers to your comments and suggestions below as authors’ responses.

I have looked at the Abstract, and even after a short review, it appears that the points raised have not yet been corrected.

- For example, a first-time abbreviation should indicate what it stands for (what is SPS, EPS?).

Authors Response: Sorry,  you are correct, we have addressed them in line 34 to 46, but forgot to add them in our last version of the abstract – they have been added now.

-Also, there is a mix of references with and without names in front of the numbers. It should be unified.

Please see track changes

Authors Response: The reason for this is because when we are paraphrasing from a reference  we put the reference at the end without the name as requested. When we are stating what someone said, we have to put their names before the reference number – We tried to follow the correct grammar. However, we agree this does not look like a unified structure or methodology, therefore we put the in text reference at the end of the sentence.

Hope that this is in line with what is expected.

Thank you

The authors 

Reviewer 2 Report

The revised version of the manuscript can be accepted 

Author Response

Thank you for the very valid comments and suggestions, which we have agreed to and incorporated in our paper. These suggestions and comments have made our paper flow much better and gave it more stability and strength in its value.  

Thank you again

The authors